# Evolution of Photorespiratory Glycolate Oxidase among Archaeplastida

**DOI:** 10.3390/plants9010106

**Published:** 2020-01-15

**Authors:** Ramona Kern, Fabio Facchinelli, Charles Delwiche, Andreas P. M. Weber, Hermann Bauwe, Martin Hagemann

**Affiliations:** 1Plant Physiology, Institute of Biological Sciences, University of Rostock, Albert-Einstein-Str. 3, D-18051 Rostock, Germany; ramona.kern@uni-rostock.de (R.K.); hermann.bauwe@uni-rostock.de (H.B.); 2Institute of Plant Biochemistry, Cluster of Excellence on Plant Science (CEPLAS), Heinrich-Heine-University Düsseldorf, 40225 Düsseldorf, Germany; fabio.facchinelli@uni-duesseldorf.de (F.F.); andreas.weber@uni-duesseldorf.de (A.P.M.W.); 3Department of Cell Biology and Molecular Genetics, University of Maryland, College Park, MD 20742, USA; delwiche@umd.edu; 4Department Life, Light & Matter, University of Rostock, 18051 Rostock, Germany

**Keywords:** Glycolate oxidase, photorespiration, evolution, Archaeplastida, Cyanobacteria

## Abstract

Photorespiration has been shown to be essential for all oxygenic phototrophs in the present-day oxygen-containing atmosphere. The strong similarity of the photorespiratory cycle in cyanobacteria and plants led to the hypothesis that oxygenic photosynthesis and photorespiration co-evolved in cyanobacteria, and then entered the eukaryotic algal lineages up to land plants via endosymbiosis. However, the evolutionary origin of the photorespiratory enzyme glycolate oxidase (GOX) is controversial, which challenges the common origin hypothesis. Here, we tested this hypothesis using phylogenetic and biochemical approaches with broad taxon sampling. Phylogenetic analysis supported the view that a cyanobacterial GOX-like protein of the 2-hydroxy-acid oxidase family most likely served as an ancestor for GOX in all eukaryotes. Furthermore, our results strongly indicate that GOX was recruited to the photorespiratory metabolism at the origin of Archaeplastida, because we verified that Glaucophyta, Rhodophyta, and Streptophyta all express GOX enzymes with preference for the substrate glycolate. Moreover, an “ancestral” protein synthetically derived from the node separating all prokaryotic from eukaryotic GOX-like proteins also preferred glycolate over l-lactate. These results support the notion that a cyanobacterial ancestral protein laid the foundation for the evolution of photorespiratory GOX enzymes in modern eukaryotic phototrophs.

## 1. Introduction

Oxygenic photosynthesis is among the most important biological processes on Earth, because it produces the vast majority of organic carbon and nearly all of the atmospheric oxygen. This process is thought to have evolved approximately 2.5 billion years ago in cyanobacteria, and was later conveyed into a eukaryotic host cell via endosymbiosis, giving rise to plastids [1,2,3]. The phototrophic eukaryotes Glaucophyta, Rhodophyta (red algae), and Viridiplantae (green algae and land plants) form the monophyletic group of the Archaeplastida, which all harbor primary plastids that evolved from a common cyanobacterial ancestor [4] (for an alternative hypothesis of multiple origins and convergent evolution of primary plastids, see [5]). Other eukaryotic algal groups contain plastids, which evolved via secondary- or higher-order endosymbiosis of a green or red algal ancestor [4]. The green lineage (Viridiplantae) comprises two major clades, the “classical” green algae (Chlorophyta), and their sister lineage, the charophyte green algae, from which land plants evolved (Streptophyta) [6,7]. 

All organisms performing oxygenic photosynthesis fix inorganic carbon as CO_2_ by Rubisco (Ribulose 1,5-bisphosphate carboxylase). In addition to the carboxylation of ribulose 1,5-bisphosphate (RuBP) with CO_2_, Rubisco also catalyzes the oxygenation of RuBP, being the first step in a process termed “photorespiration”, leading to the appearance of the potent enzyme inhibitor 2-phosphoglycolate (2PG) [8,9]. Despite the evolution of different types of Rubisco with varying substrate affinities and specificity factors, the oxygenase reaction cannot be avoided under the present atmospheric conditions, containing 21% O_2_ and 0.04% CO_2_ [10]. As 2PG is toxic for the plant’s metabolism [11,12,13], it is rapidly metabolized to glycolate, which is efficiently metabolized in the photorespiratory pathway [14]. However, this salvage process is energetically costly and recovers only 75% of the organic carbon, while 25% is lost as CO_2_. It has been estimated that in a crop, C3 plant photorespiration might decrease the yield by approximately 30% under present day atmospheric conditions [15]. Thus, photorespiration is one key target in molecular breeding attempts to improve crop productivity [16,17].

Initially, it was assumed that photorespiration evolved rather late when plants started to colonize the continents and became exposed to the high O_2_-containing atmosphere with low CO_2_ (e.g., [18]). In contrast, we hypothesized that photorespiration and oxygenic photosynthesis coevolved in ancient cyanobacteria [14,19]. This hypothesis is based on the discovery of the essential function of the photorespiratory 2PG metabolism under ambient air conditions in the model cyanobacterium *Synechocystis* sp. PCC 6803 [19]. However, the cyanobacterial photorespiratory 2PG metabolism differs from that of plants. First, it is not compartmentalized in the prokaryotic cell. Second, glycolate oxidation is performed by glycolate dehydrogenases among cyanobacteria, and not by glycolate oxidase (GOX), as in plants (Figure 1). Third, in addition to a plant-like 2PG metabolism, *Synechocystis* can also metabolize glyoxylate via the bacterial glycerate pathway and by its complete decarboxylation [19]. The subsequent phylogenetic analysis also revealed that not all enzymes of the plant 2PG cycle originated from the cyanobacterial ancestor. In particular, the photorespiratory enzymes that operate in the mitochondrion of plants are more closely related to enzymes of proteobacteria, the ancestors of this organelle [20,21]. We hence hypothesized that these enzymes, which originally also existed in the cyanobacterial endosymbiont, were replaced by the already established mitochondrial enzymes of a proteobacterial origin [22].

However, in contrast to plants and similar to cyanobacteria, chlorophyte algae, such as *Chlamydomonas reinhardtii*, also use glycolate dehydrogenase for glycolate oxidation, not GOX [23]. The presence of glycolate dehydrogenases in green algae and the later discovery of similar enzymes in cyanobacteria correlated with the existence of an efficient inorganic carbon concentrating mechanism (CCM), which suppresses photorespiration to a great extent in these organisms. Therefore, it was initially assumed that cyanobacteria and chlorophytes with a lower photorespiratory flux due to the CCM prefer glycolate dehydrogenases, which have a higher affinity to glycolate than GOX [24,25]. Accordingly, it was proposed that plant-like photorespiration, including the peroxisomal glycolate oxidation via GOX, only evolved late among streptophytic green algae, and represented an adaption to a low CO_2_ and high O_2_ concentration, which was indeed necessary for the later terrestrialization of the plant kingdom [18].

Recently, numerous new genome sequences became publicly available, which permitted a broader analysis of the distribution of GOXes and glycolate dehydrogenases among different algal lineages. Those searches also revealed that some cyanobacterial genomes possess a GOX-like protein [20]. The subsequent biochemical analysis of the GOX-like protein from the diazotrophic cyanobacterium *Nostoc* sp. PCC 7120 showed that this protein is rather a l-lactate oxidase (LOX). LOX and GOX (here summarized as GOX-like proteins) belong to the group of 2-hydroxy-acid oxidases, display highly similar primary and tertiary structures (see Appendix A), and share the same catalytic mechanism [26,27,28,29]. Phylogenetic analysis using GOX-like proteins from heterotrophic bacteria, cyanobacteria, eukaryotic algae, and plants implied that the cyanobacterial GOX-like protein is the common ancestor of all plant GOX proteins, which was consistent with the hypothetical transfer of genes for photorespiratory enzymes into the plant genome via primary endosymbiosis from the previously assumed ancestral N_2_-fixing cyanobacterium [30]. However, more recent phylogenetic analyses point at the likely uptake of a non-N_2_-fixing, early-branching unicellular cyanobacterium as a primary endosymbiont [31]. Another study included GOX-like proteins from non-photosynthetic eukaryotes in the phylogenetic analysis, and postulated that plant and animal GOX proteins share a common non-cyanobacterial ancestry [32].

Hence, to solve this controversy and better understand the evolutionary origin of the photorespiratory GOX, we reanalyzed GOX-like proteins using biochemical and phylogenetic approaches. Our results support the view that GOX became part of the photorespiratory metabolism early in the evolution of Archaeplastida, and that a cyanobacterial GOX-like protein most likely served as the ancestor for GOX in all eukaryotic lineages.

## 2. Results

### 2.1. Data Mining

To evaluate the GOX phylogeny, we considered a broad spectrum of phyla, including non-photosynthetic species. The well-characterized At-GOX2 from *Arabidopsis thaliana* (At3g14415) [30] was used in BLASTP [33] searches against defined taxonomic groups (e.g., cyanobacteria or proteobacteria) to find related GOX-like proteins. Proteins from 5 to 14 species of each taxonomic group were selected, which showed the best BLAST hits. In the case of underrepresented taxonomic groups, all of the sequences that could be identified as putative GOX proteins were used for the alignment (Appendix A). In addition to the sequences in the databases, we determined the cDNA sequence of the putative GOX from the streptophyte green alga *Spirogyra pratensis* (sequence is shown in Appendix A; accession number AVP27295.1). Furthermore, the obviously miss-annotated GOX sequence of *Cyanophora paradoxa* was corrected (see below). The complete gene with a corrected exon/intron structure and the complete protein coding sequence is shown in Appendix A (Appendix A; accession number AVP27296.1).

### 2.2. Monophyly of Eukaryotic and Cyanobacterial GOX-Like Proteins 

To reanalyze the phylogenetic origin of the plant and animal GOX-like proteins, we took sequences from a broad spectrum of phyla into consideration, for example, we also included related proteins from the chromalveolate taxa (Appendix A). In total, 111 GOX-like proteins from 11 groups, including Archaeplastida and Metazoa, were incorporated into the phylogenetic analysis. GOX-like proteins from fungi were excluded, as these proteins show an accelerated evolution preventing their comparison [32]. We also restricted the analysis to one putative GOX isoform each from algae, plants, and animals, as the previous study [32] showed that the diversification of GOX into different biochemical subgroups occurred from one of the ancestral proteins within these groups. The alignment was constructed with ProbCons [34]. Additionally, we also constructed alignments using MUSCLE [35], compared both results, and changed the alignment if necessary in order to obtain the best scores. Using the final alignment (Appendix A), we reconstructed a protein tree using Bayesian interference. 

The midpoint-rooted Bayesian tree (Figure 2) is well supported by the Bayesian posterior probabilities (BPP). The GOX-like proteins from all of the eukaryotes form a monophyletic group (BPP = 0.96). The proteins from Chlorophyta cluster in that group, however, they build an outgroup to other eukaryotic GOX-like proteins and are distinct from that of other Archaeplastida. The divergence of the GOX proteins of streptophytes is consistent with the fact that the chlorophyte GOX-like proteins act as LOX enzymes (see below). The GOX-like sequences from the Metazoa cluster together with sequences of chromalveolate taxa, including non-photosynthetic and photosynthetic groups; this clade is the sister clade to all Archaeplastida, except Chlorophyta (BPP = 1). Interestingly, LOX proteins from cyanobacteria cluster as sisters to all eukaryotes, showing a close relationship between cyanobacterial and eukaryotic GOX-like proteins. It is also noteworthy that the GOX-like proteins of chromalveolates do not cluster within red algae, as would be expected from the secondary origin of their plastids, but rather form a clade with Metazoa, although the posterior probabilities for the critical branches placing red algae are relatively low. The cyanobacterial and eukaryotic clades of GOX-like proteins are separated from the proteins of Actinobacteria (also including non-sulfur bacteria), as well as from the clade Proteobacteria (also including Verrucomicrobia; BPP = 1). The outermost clade is built by Firmicutes and Archaea. A similar clustering of GOX-like proteins was obtained using the maximum likelihood (ML) algorithm (Appendix A). However, in contrast to the Bayesian tree, some branches are not well supported in the ML tree, but all cyanobacterial and eukaryotic GOX-like proteins again form one monophyletic, statistically well-supported clade.

Similar to the previous study by Esser et al. [32], we also rooted the tree by assuming the monophyly of the eukaryotic clade (Appendix A). This rooting resulted in a monophyletic group of all Eubacteria and Archaea, which cluster in a sister group relationship to Eukaryotes, including plants, algae, and animals. However, the clustering of the bacterial and archaeal GOX-like proteins in this rooted tree is not congruent to the clustering of taxonomic groups, which is observed when analyzing, for example, signature sequences in proteins [36]. In both trees, Archaea and Firmicutes cluster as sister groups. Alternatively, we placed the root of the tree between Firmicutes or Archaea and the other clusters of GOX-like proteins (Appendix A). Both of these trees again showed a sister group topology of sequences from eukaryotes and cyanobacteria, as found before with the midpoint-rooted tree.

Overall, our phylogenetic analyses show that eukaryotic GOX-like proteins are more closely related to those of cyanobacteria than to those of any other prokaryotic group, including Proteobacteria and Archaea. As it is known that phylogenetic analysis can be biased when homologous proteins evolved into enzymes with varied metabolic functions, we set out a comprehensive biochemical evaluation of GOX-like proteins among Archaeplastida.

### 2.3. Archaeplastida Except Chlorophyta Possess a GOX Protein

It is known that GOX-like proteins show varying (multi-)substrate preferences. For example, many of these enzymes preferentially oxidize l-lactate, that is, they rather represent LOX than GOX enzymes. Therefore, we analyzed the substrate preference of GOX-like proteins from all groups of Archaeplastida. To this end, we overexpressed selected cDNAs in *E. coli* to obtain recombinant proteins for biochemical characterization. In addition to the previously obtained data for GOX and LOX proteins from Cyanobacteria, Rhodophyta, Chlorophyta, and land plants [30,38], the His- or Strep-tagged proteins from the early splitting-off glaucophyte alga *Cyanophora paradoxa* and the streptophyte (sister clade of land plants) alga *Spirogyra pratensis* were biochemically characterized using substrate concentrations of glycolate and l-lactate ranging from 0.1 to 200 mM. We found that both recombinant proteins can oxidize glycolate and, to a lesser extent, l-lactate (Figure 3 and Table 1), which is consistent with earlier reports of clear GOX activity in crude extracts from these algae [24,39,40]. Compared to At-GOX2 from *Arabidopsis*, both enzymes show a higher affinity (at least 10 times) to glycolate than to l-lactate as a substrate. Although the Vmax values are only slightly higher for glycolate than for l-lactate, the catalytic efficiency (k_cat_/K_m_) is 20 to 33 time higher with glycolate as substrate, which is similar to AtGOX2 (Table 1 and Appendix A). Based on this preference for glycolate as substrate, the enzymes were named Cp-GOX and Sp-GOX, respectively. Compared with other GOX proteins, Cp-GOX shows the lowest K_m_ and the second highest k_cat_/K_m_ value for glycolate (Table 1 and Appendix A). This finding could point to an early evolution of glycolate oxidation activity in the ancestor of cyanobacteria. To verify this hypothesis, we modeled, synthesized, and characterized an ancestral GOX protein. 

### 2.4. Ancestral GOX-Like Protein Sequence with Active Site Identical to Plant GOX

In contrast to the proteins of Archaeplastida, except Chlorophyta, cyanobacteria (Table 1) and all other analyzed prokaryotes possess LOX, a GOX-like protein that prefers l-lactate over glycolate [27,30,41]. To get an idea about the substrate preference of an ancestral protein, we reconstructed a protein that could correspond to the hypothetical common ancestor of GOX-like proteins from Archaeplastida. The primary structure of this “ancestral” protein was derived from a reduced phylogenetic tree based on Bayesian interference, including 37 sequences from five different taxonomic groups (Figure 4). Also, for the reduced dataset, the monophyly of cyanobacterial and eukaryotic GOX-like proteins is supported by a high posterior probability (BPP = 1). The ancestral protein, named N3-GOX, used for the biochemical analysis referred to the most probable amino acid sequence calculated via the ML algorithm. An amino acid sequence comparison of N3-GOX with At-GOX2 from *Arabidopsis* revealed that the three amino acid residues in the active site (Table 2), which were previously proven to determine the specificity for the substrate glycolate [30], were identical in these enzymes, pointing to a higher GOX rather than LOX activity of the ancestral protein. Comparing the entire sequence, 68% of the amino acid residues are identical in these two proteins. In contrast, the cyanobacterial No-LOX (from *Nostoc* sp. PCC7120 [30]) shares only 56% identical positions with N3-GOX, and shows different active-site amino acid residues (Table 3). 

### 2.5. Activity of Ancestral GOX Proteins Point to Early Evolution of Preferential Glycolate Oxidation

To analyze the enzymatic activity of the ancestral N3-GOX protein, which represents a proxy for the common ancestor of GOX-like proteins from Archaeplastida, the gene was synthesized, cloned, and expressed in *E. coli*. The subsequent biochemical analysis was done as described for the extant GOX-like proteins, using glycolate and l-lactate concentrations ranging from 0.2 to 100 mM and 0.5 to 200 mM, respectively. The “ancestral” N3-GOX showed a preference for glycolate as a substrate, as reflected in a higher V_max_ and k_cat_/K_m_ value for glycolate (Figure 5 and Appendix A). However, compared with the extant proteins At-GOX2 and Sp-GOX from the streptophyte clade, N3-GOX shows a 30 to 35 times lower V_max_ (0.87 µmol min^−1^ mg^−1^ protein) and a slightly higher k_cat_/K_m_ with glycolate as a substrate (Figure 5, Table 1, and Appendix A). Although the K_m_ value for glycolate is lower, the difference is not significant, which is in contrast to most of the other GOX-like proteins [30,38]. While the proteins from *Cyanidioschyzon* and *Spirogyra* show a similar affinity for l-lactate, the affinity for glycolate is at least 10 times lower in the “ancestral” N3-GOX compared with the extant GOX and LOX proteins (Table 1).

## 3. Discussion

GOX-like proteins belong to the (S)-2-hydroxy-acid oxidase protein family. Besides the important role of GOX in the photorespiration of plants, where it oxidizes glycolate to glyoxylate (Figure 1; e.g., [42]), GOX-like enzymes are also present in animals and heterotrophic prokaryotes, where they can also metabolize other hydroxy acids such as l-lactate, and are additionally involved in fatty acid metabolism or in the detoxification of critical intermediates [43,44]. Despite the differences in their substrate preference [30,32,43,45], all of these enzymes are clearly members of a single gene family and share similar homo-tetrameric quaternary structures (see Appendix A) [46,47].

Their wide-spread occurrence in almost all bacterial and eukaryotic groups makes the emergence of the eukaryotic GOX-like proteins from a bacterial ancestor very likely. Indeed, our phylogenetic analysis (Figure 2) indicates that all eukaryotic GOX-like proteins evolved from an ancestral protein of ancient cyanobacteria. This scenario includes the LOX proteins of extant cyanobacteria, which build the sister group to the clade of eukaryotic GOX-like proteins. This sister group topology is preserved when alternative roots are used for tree building (see Appendix A). Thus, the phylogenetic reconstruction supports the view that all eukaryotic GOX-like proteins most likely evolved from prokaryotic ancestors, likely acquired from ancient cyanobacteria. This view, also supported by the close biochemical and protein similarities, is more parsimonious than the assumption of an independent evolution of these proteins among eukaryotes, although it is surprising in the sense that it implies gene transfer from a cyanobacterium to a very early eukaryote. As expected, the topology of the tree shown in Figure 2 shares many features with a previously reported reconstruction [32]. However, the inclusion of important additional taxa and different rooting have significant new implications for our understanding of the evolution of GOX-like proteins.

Placing the root of the tree for GOX-like proteins in between eukaryotic and prokaryotic proteins, as was done by Esser et al. [32], unsurprisingly results in the separation of eukaryotic from all prokaryotic GOX-like proteins (see Appendix A), and an appearance as if the eukaryotic and prokaryotic proteins would have evolved independently. However, given the evidence that Eukarya are closely related to, and may in fact be derived from Archaea [48,49], the interpretation of the eukaryotic GOX-like proteins as being the native eukaryotic form is difficult to reconcile with the phylogenetic reconstructions. If eukaryotic GOX-like proteins indeed originated from the archaeal ancestor, then related GOX-like proteins should be found in at least some Archaea. However, only a few GOX-like sequences have been found among Archaea, and all of them are restricted to Euryarchaeota. In our midpoint-rooted tree (Figure 2 and Appendix A), the GOX-like proteins from Archaea cluster with the Firmicutes as the sister group to all other sequences. An analysis of the protein signatures has placed the Firmicutes as an outgroup to most other bacteria [36], although other analyses place them closer to cyanobacteria [50]. Furthermore, the rare occurrence of GOX-like proteins among Archaea and their sister-group relation with Firmicutes suggests that related proteins were present in the last universal common ancestor and lost from other Archaea, or, alternatively, have undergone a horizontal gene transfer (HGT) event. In fact, HGT is a prominent force in prokaryotic and archaeal genome evolution [51]. Placing the GOX-like proteins of Archaea as the outgroup, all bacterial and eukaryotic sequences are found in one clade, where cyanobacterial and eukaryotic proteins are found as sister groups (see Appendix A). 

Hence, our phylogenetic tree strongly implies an origin of GOX in all eukaryotes from a GOX-like protein of the 2-hydroxy-acid oxidase family of ancient cyanobacteria. Such a relation is easily understandable for eukaryotic phototrophs, which evolved because of the endosymbiotic uptake of a cyanobacterial cell that eventually formed the plastids [1]. However, the origin of the GOX-like proteins of chromalveolate taxa and Metazoa from cyanobacteria is more difficult to understand. It is widely accepted that animal (and fungal) lineages separated from the phototrophic eukaryotes before the endosymbiotic engulfment of plastids. Hence, the sister group relationship of animal GOX-like proteins with cyanobacterial proteins is not easily explainable. One possibility is that the placement of this clade is incorrect because of a phylogenetic reconstruction artifact; however, the clade is quite distant from Proteobacteria (mitochondria) and Archaea, the other expected placements for eukaryotic genes obtained from prokaryotes. It is interesting to note that a few other animal proteins appear to be of a cyanobacterial origin. For example, the animal alanine–glyoxylate aminotransferase, which is, like GOX, located in the peroxisome, appears to be derived from cyanobacteria [52]. Moreover, the animal aldehyde dehydrogenases and cytochrome P450 enzymes have their closest relatives with cyanobacterial orthologs [53]. Finally, the photorespiratory glycerate 3-kinase (GLYK) is shared by cyanobacteria, fungi, and all eukaryotic phototrophs [20,54,55]. A simple and parsimonious explanation for the unexpected cyanobacterial protein origins is an early HGT between an ancestral cyanobacterium and the common ancestor of eukaryotes. The occurrence of not only one, but several cyanobacterial proteins in the non-phototrophic eukaryotes could further indicate an earlier endosymbiotic event between a eukaryotic cell and a cyanobacterium, which, however, did not result in a stable establishment of a plastid. Thus, these cyanobacterial genes could be seen as relics from earlier transfers, possibly even ones that somehow prepared the eukaryotic host for the final successful plastid incorporation as suggested by the “shopping bag” hypothesis [56], that is, multiple endosymbiotic gene transfer. 

Our results support the inference that the photorespiratory GOX in all phototrophic prokaryotes originated from an ancestral cyanobacterial protein [30]. We verified the presence of a biochemically active GOX in glaucophyte algae, red algae [38], and streptophyte green algae (Table 1). Thus, the consistent utilization of a photorespiratory GOX among all groups of Archaeplastida points to an early evolution of the plant-like photorespiratory cycle in the common ancestor of phototrophic eukaryotes. This notion is also supported by the biochemical analysis of the reconstructed ancestral N3-GOX, which displays some preference of glycolate over l-lactate as a substrate. Previously, it was assumed that the photorespiratory glycolate oxidation via GOX only appeared in Streptophyta, because chlorophytes such as *Chlamydomonas* perform this reaction by a mitochondrion localized glycolate dehydrogenase [57,58,59]. The genome sequence of *Chlamydomonas* revealed that, in addition to glycolate dehydrogenase, this chlorophyte also possesses a GOX-like protein. However, biochemical analyses showed that, in contrast to the GOX of all other Archaeplastida, the *Chlamydomonas* enzyme must be assigned as LOX, as it clearly prefers the substrate l-lactate over glycolate, similar to the GOX-like proteins from N_2_-fixing cyanobacteria such as *Nostoc* (Table 1) [30]. 

The close similarity of cyanobacterial and chlorophyte LOX proteins is also reflected in our phylogenetic tree, where the chlorophytic LOX proteins form an extra clade at the base of all eukaryotic GOX-like sequences, clearly separated from the GOX clade of all remaining Archaeplastida. There are two possible scenarios how LOX evolved among chlorophytes. The most likely explanation is that the LOX in Chlorophyta reflects a functional reversal from an early GOX; this may reflect a clade that diverged before the ancestral cyanobacterial 2-hydroxy-acid oxidase diversified among Archaeplastida, but may also reflect a phylogenetic artifact stemming from differential selection, depending on the substrate. As mentioned before, all GOX-like enzymes can use l-lactate and glycolate to some degree, and it is to be expected that over the course of evolutionary history, they were optimized to fit the specific metabolic requirements of the respective organism as GOX or LOX. A second possible explanation is that an ancient cyanobacterial LOX received by the endosymbiotic uptake of the plastid ancestor was retained as a second gene in the chlorophyte genome, while it was lost from other algal genomes, or has been obtained by a second HGT from cyanobacteria, as suggested by Esser et al. [32]. Regardless of which scenario actually occurred, LOX in chlorophytes also originated from a protein closely related to those of cyanobacteria. 

Furthermore, the sequence and biochemical analyses of the GOX-like proteins among Archaeplastida and cyanobacteria revealed that the experimentally verified amino acid residues that determine whether glycolate or l-lactate is the preferred substrate [30] also distinguish the cyanobacterial and chlorophyte LOX from GOX, such as the Cp-GOX from *Cyanophora*, the Cm-GOX from *Cyanidioschyzon*, and the streptophytic Sp-GOX from *Spirogyra* (Table 2). Interestingly, the hypothetical ancestral N3-GOX also possesses the glycolate-preferring amino acid signature, corresponding to its dominant enzymatic activity as GOX. The amino acid residues responsible for the binding of the flavin mononucleotide (FMN) cofactor are also highly conserved in all GOX and LOX proteins, analyzed here or previously [30,38]. Interestingly, we found two clear trends among the biochemically verified GOX enzymes among Archaeplastida (Table 1). The GOX of the early branching Glaucophyta showed the highest affinity for glycolate among phototrophic eukaryotes, whereas streptophyte GOX proteins have a five-times lower affinity for glycolate. The inverse trend is observed regarding the V_max_ values, which are the highest among Streptophytes and the lowest for the enzyme from Glaucophyta. These findings are consistent with the hypothesis that Streptophytes, especially C3 plants, are characterized by much higher photorespiratory fluxes compared with algae, in which photorespiration is often inhibited by the presence of a CCM. Thus, a highly active peroxisomal GOX allows for the rapid degradation of glycolate and its recycling to 3-phosphoglycerate [24,60]. The relatively low V_max_ of the N3-GOX may also result from incorrectly predicted amino acid residues, which can lead to protein misfolding [61].

## 4. Materials and Methods 

### 4.1. Phylogenetic Analysis

The selected amino acid sequences of all GOX-like proteins were aligned using ProbCons [34], which produces the best alignment scores [62]. An alternative alignment was constructed using MUSCLE [35]. Both alignments, especially their non-ambiguous parts, were compared via the program SuiteMSA [63]. To make sure that the incorrect aligned position was not considered for the phylogenetic analyses, we used the alignment curation program GBlocks [64]. Nearly identical sequences (after Gblocks curation) or sequences with a different evolutionary rate (calculated via Tajima’s rate test included in MEGA 5.1; [65]) were excluded from further analyses. We also excluded all sequences from single species, which did not cluster into the clade of their taxonomic group to minimize HGT influences. ProtTest 3 [66] was used to find the best fitting substitution model for the given data set. The model of Le and Gascuel [67], including a proportion of invariant sites and a gamma model of rate heterogeneity, was found to be the most appropriate model of evolution for the alignment. For the tree reconstruction, we used the Bayesian approach implemented in MrBayes [68], running 2,000,000 generations and disregarding the first 25% of samples as burn-in. The likelihoods and tree topologies of the independent runs were analyzed and compared. In addition to the Bayesian approach, we also performed a maximum likelihood analysis using RAxML version 8 [69], with a bootstrap test of the statistical support from 1000 replicates.

### 4.2. Ancestral Sequence Reconstruction

For the reconstruction of the ancestral eukaryotic GOX protein sequence, we first reduced the number of sequences used for the phylogeny (shown in Figure 2) to 36 species, which are situated around the nodes of the eukaryotic GOX-like proteins. The phylogenetic analysis of the reduced dataset was performed as described above, and is shown in Figure 4. Because of the long evolutionary distance of the analyzed proteins, ancestral reconstruction based on amino acid inference were performed using CODEML included in PAML [70]. In addition, we used the FASTML program including the marginal and joint method of ancestral sequence reconstruction [71]. We also performed a Maximum Likelihood and Maximum Parsimony indel reconstruction, which provided the same results. We focused on the last common ancestor of all eukaryotes (Node 3). The deduced amino acid sequences of the Node 3 ancestral GOX-like protein was translated into a DNA sequence applying the codon-usage of *E. coli* (sequences are displayed in the Appendix A). The codon-optimized gene for the ancestral GOX protein was synthesized (Thermo Fisher, Darmstadt, Germany). For cloning into the expression vector pASG-IBA43+, the restriction sites *Nhe*I and *Bam*HI were added at the beginning and the end of the ancestral gene, respectively. 

### 4.3. Sequencing of the cDNA Encoding the GOX from Spirogyra Pratensis

The complete sequence of the *Spirogyra* GOX was extracted from an existing Expressed sequence tag (EST) library [72], which was screened via BLASTX using the *Arabidopsis* At-GOX2 (At3g14415) sequence. The full-length cDNA sequence from *Spirogyra* (Appendix A) was compared to known GOX and LOX proteins, and checked for completeness. The sequence of *Spirogyra* GOX (Sp-GOX) was deposited in GenBank under the accession number AVP27295.1.

### 4.4. Estimation and Synthesis of the cDNA Encoding the GOX from Cyanophora Paradoxa

The GOX sequence from *C. paradoxa* was identified by BLAST searches with plant and algal GOX proteins. The originally annotated *Cyanophora* protein of the genomic Contig54585 consists of 316 amino acids, which is considerably shorter than the GOX protein from *Arabidopsis* (367 amino acids). Initially, we used this cDNA to synthesize the *Cyanophora gox* gene; however, we failed to obtain an enzymatic active protein, most likely because of the missing C-terminus. Subsequent BLAST searches including EST sequences identified two different EST sequences (EG947183.1 and ES232585.1), which partially overlap with the *gox* gene in the *Cyanophora* genome database (Contig54585; http://cyanophora.rutgers.edu/cyanophora/home.php). The newly obtained cDNA for the *Cyanophora gox* gene (348 amino acids long, sequence called Cp-GOXb) was also used for gene synthesis (Thermo Fisher), but we again failed to produce an enzymatic active protein. Finally, we re-sequenced the entire *gox* gene from *Cyanophora* gDNA using new primer sets (CpGOXb_275_fw, CpGOXb_369_fw, and CpGOXb_655_rv; Appendix A). The new sequence revealed a missing 15-nucleotides-long insertion in the originally annotated genomic version of this *gox* gene. The final cDNA sequence gave rise to the corrected protein, which was named Cp-GOXc. The entire gene was synthesized by PCR using the DNA of the previously obtained expression vector pASG-IBA43plus-*cp*-*goxb* as a template and the phosphorylated primers CpGOX_RV_5P and CpGOX_FW_5P (Appendix A). The previously 15 missing bases coding for five amino acids were included via the used primer. After the ligation of the final PCR product, we obtained the expression vector pASG-IBA43plus-*cp*-*goxc* harboring the correct cDNA sequence of the *gox* gene from *Cyanophora*. A scheme including all steps leading to the corrected *gox* gene from *Cyanophora* is shown in Appendix A.

### 4.5. Cloning, Heterologous Expression and Purification of Recombinant GOX Proteins

The obtained *cp*-*gox* gene was subsequently cloned without a stop codon into the expression vector pASG-IBA43plus. The resulting strep-tagged fusion proteins were generated in *E. coli* strain BL21 GOLD and purified using the Strep-tactin matrix, according to the supplier’s protocol (IBA Bio technology, Göttingen, Germany). The *gox* gene of *Spirogyra* and the gene encoding the ancestral GOX protein were also cloned into pASG-IBA43plus. However, the N-terminal His-tag was used for purification with Ni-NTA Sepharose, according to the supplier’s protocol (Invitrogen, Karlsruhe, Germany). 

For the production of Sp-GOX and Cp-GOX, the recombinant *E. coli* strain was grown in an LB medium to an OD_750_ of 0.6. The gene expression was initiated with the addition of 200 µg/L anhydrotetracycline for 4 to 16 h at 20 to 30 °C. A soluble protein of the ancestral N3-GOX could be only obtained when cultivating *E. coli* cultures at 18 °C for 16 h after induction with anhydrotetracycline. *E. coli* cells were harvested by centrifugation and re-suspended in buffer A (20 mM Tris-HCl, 500 mM NaCl, 1 mM dithiothreitol (DTT), and 0.1 mM Flavin mononucleotide (FMN)) in case of His-tag fusion proteins or buffer W (100 mM Tris pH 8.0, 150 mM NaCl, 1 mM EDTA, 1 mM DTT, and 0.1 mM FMN) in case of Strep-tag proteins. The protein extraction was done by ultrasonic treatments (4 times 30 s, 90 W) on ice. After centrifugation (20,000 g, 20 min, 4 °C, Sorvall SS34 rotor), the cell-free protein extract was directly loaded onto Ni-NTA or Strep-tactin columns. The His-tagged proteins were washed using buffer A supplemented with 40 to 80 mM imidazole, whereas His-tagged GOX was eluted with buffer A supplemented with 200 mM imidazole. For Strep-tagged proteins, the washing steps were performed using the buffer W, and the proteins were eluted by buffer W, supplemented with 2.5 mM desthiobiotin. Subsequently, the eluted proteins were desalted using PD-10 columns (GE Healthcare, Schwerte, Germeny), and finally dissolved in 20 mM Tris/HCl pH 8.0 containing 1 mM DTT and 0.1 mM FMN. The purity of the eluted proteins was checked by SDS-PAGE and staining with Coomassie Brilliant Blue (Appendix A). 

### 4.6. Enzyme Activity Assays 

The GOX or LOX activity was measured by the detection of O_2_ consumption in the presence of different concentrations of l-lactate and glycolate using Hansatech oxygen electrodes (Oxygraph), as described by Hackenberg et al. [30]. One unit of enzyme activity was defined as the consumption of 1 µmol O_2_ in 1 min at 30 °C. The protein concentrations were estimated according to the literature [73]. 

## 5. Conclusions

The monophyly of eukaryotic and cyanobacterial GOX-like proteins points to a cyanobacterial origin of all eukaryotic GOX-like proteins, including the photorespiratory GOX of phototrophic eukaryotes. Biochemically, these proteins operate as GOX rather than LOX in the Archaeplastida. The slight preference for glycolate of the synthetic ancestral N3-GOX suggests that glycolate oxidation could have been the ancestral function, and during the course of evolution, this protein has changed its preference to glycolate or l-lactate and other related substrates according to the requirements of the specific host. 

The likely scenario for the evolution of GOX among eukaryotes and particularly of the photorespiratory GOX of Archaeplastida is summarized in Figure 6. A bifunctional GOX-like protein (represented by N3-GOX) was transferred from an ancient cyanobacterium via HGT into eukaryotes before the split of animal, fungal, and plant lineages occurred. Most extant cyanobacteria lost this enzyme, and use a glycolate dehydrogenase for photorespiratory glycolate oxidation, whereas N_2_-fixing cyanobacteria optimized the ancestral GOX-like protein to operate as LOX, which helps consume oxygen, protecting nitrogenase [30]. Among the Metazoa, the ancestral cyanobacterial protein evolved into GOX-like enzymes with varying substrate specificities after gene duplication and biochemical specialization, as suggested by Esser et al. [32]. After the engulfment of an ancient cyanobacterium as a plastid ancestor, possibly two gene copies for GOX-like proteins co-existed in the proto-alga, but only one of these copies was retained. Among the Archaeplastida, the ancestral GOX-like protein early on evolved to the main photorespiratory glycolate oxidizing enzyme and became localized in peroxisomes. The evolution of peroxisomes in eukaryotes is still a matter of discussion, but it has been shown that the proteome of peroxisomes is variable, pointing to an evolutionary optimization of peroxisome functions in time by protein acquisitions and losses [74]. The increasing photorespiratory flux due to the increasing oxygen concentration in the environment was matched by increasing the V_max_ of glycolate oxidation.

Hence, the early acquisition of a peroxisomal GOX-like protein by the hypothetical host cell [75,76,77] can be regarded as further exaptation to integrate the photosynthetic plastid ancestor with an already active host glycolate metabolism in the oxygen-containing atmosphere [22].

## Figures and Tables

**Figure 1 plants-09-00106-f001:**
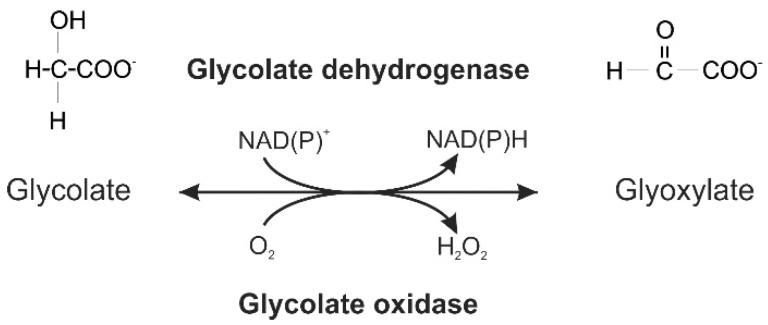
Glycolate oxidation can be catalyzed either by a glycolate dehydrogenase (top) or a glycolate oxidase (bottom).

**Figure 2 plants-09-00106-f002:**
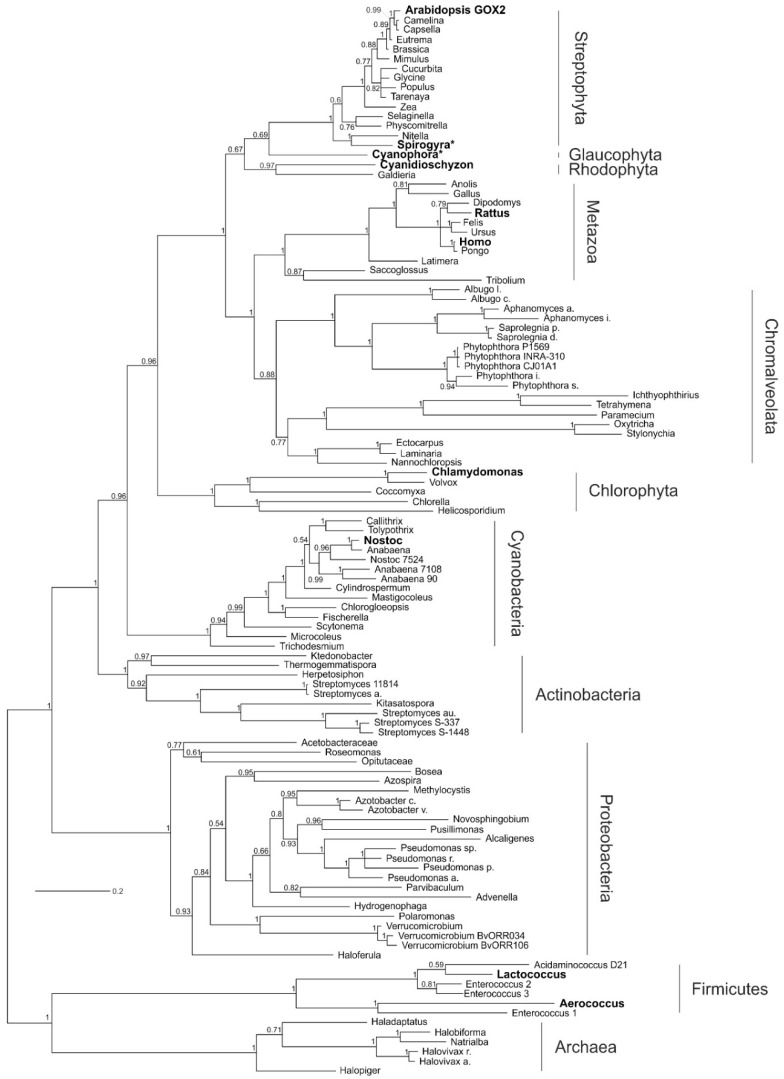
Phylogenetic tree of glycolate oxidase (GOX)-like proteins. The tree is based on GOX-like proteins from all groups in the tree of life and was implemented using Bayesian interference. The monophyletic group of GOX-like proteins from Eukaryotes builds the sister clade to cyanobacteria, pointing to a cyanobacterial origin of all eukaryotic GOX proteins, including heterotrophic organisms like animals. GOX-like proteins that have been biochemically verified are printed in bold (https://www.brenda-enzymes.org/index.php; EC 1.1.3.15; [37]). Proteins with a star were analyzed in this study. The numbers at the nodes show the posterior probability. The full species names and accession numbers are listed in Appendix A. GOX—glycolate oxidase; LOX—lactate oxidase.

**Figure 3 plants-09-00106-f003:**
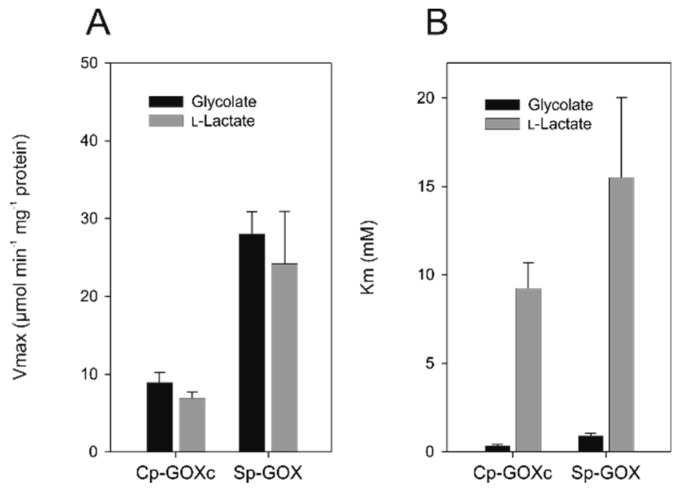
Biochemical characterization of GOX proteins from algae. The nonlinear regression was fitted to the Michaelis–Menten kinetic using Sigma Plot 13.0. The resulting parameters for the maximal enzymatic reaction rate (V_max_) (**A**) and the substrate affinity expressed as K_m_ (**B**) were calculated for the substrates glycolate and l-lactate. The standard deviation was calculated from at least two independent biological replicates. Cp-GOXc—*Cyanophora paradoxa*; Sp-GOX—*Spirogyra pratensis*.

**Figure 4 plants-09-00106-f004:**
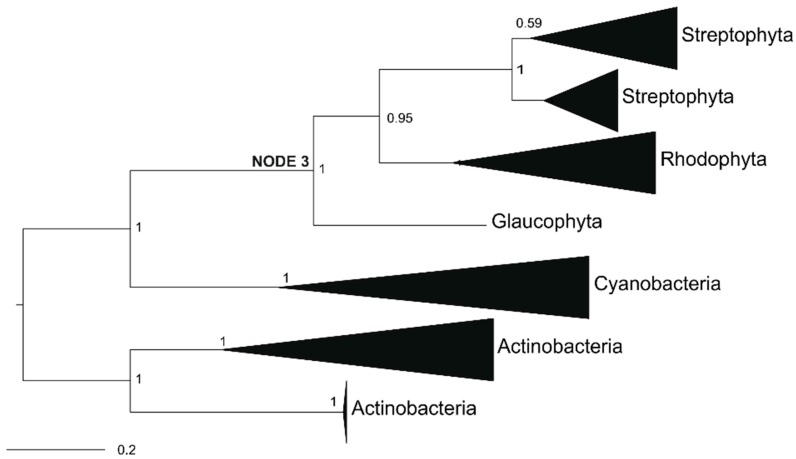
Phylogenetic tree of GOX-like proteins used for “ancestral” protein sequence reconstruction based on Bayesian interference. The numbers at nodes show the posterior probability. The full species names of all of the summarized groups can be found in Appendix A. The derived amino acid for Node 3 is the ancestral sequence of eukaryotic GOX-like proteins.

**Figure 5 plants-09-00106-f005:**
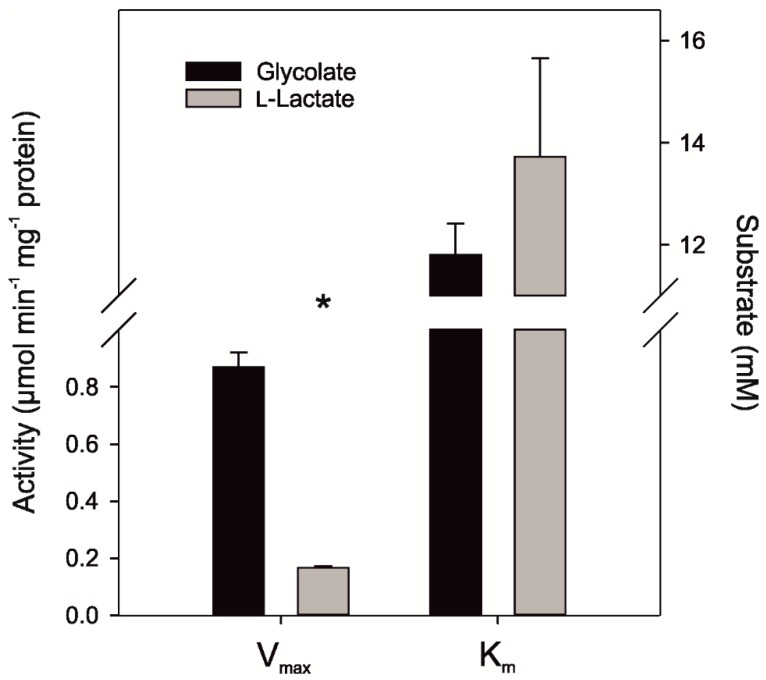
Biochemical characterization of the synthetic “ancestral” N3-GOX protein. The K_m_ and V_max_ values for glycolate and l-lactate were calculated by non-linear regression implemented in Sigma Plot 13.0 using the Michaelis–Menten kinetic model. For each substrate, three biological replicates were used. Asterisk indicates the significant difference (*p* < 0.05) determined with the two-tailed Student’s *t*-test.

**Figure 6 plants-09-00106-f006:**
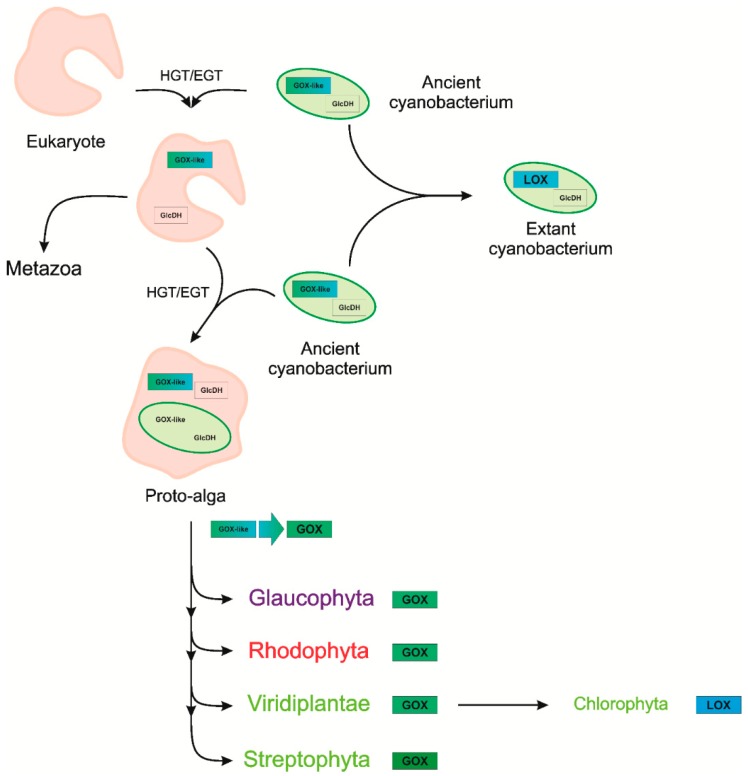
Hypothetical evolutionary scenario of GOX-like proteins of the 2-hydroxy-acid oxidase family among Eukaryotes. A bifunctional GOX-like protein and a glycolate dehydrogenase (GlcDH) existed in ancient cyanobacteria. In most extant cyanobacteria the GOX-like gene was lost, and they use GlcDH for photorespiratory glycolate oxidation. The cyanobacterial gene of the GOX-like protein was initially transferred to the eukaryotic genome via horizontal (endosymbiotic) gene transfer (HGT/EGT). After the engulfment of an ancient cyanobacterium as a plastid ancestor, probably two gene copies for GOX-like proteins existed. Subsequently, one of these copies was lost during plastid establishment. The early evolution of the glycolate specificity of the GOX-like protein most likely took place in the proto-alga before the split-off of Archaeplastida lineages. Only among cyanobacteria and chlorophytes, did GOX-like proteins evolve into l-lactate oxidase (LOX).

**Table 1 plants-09-00106-t001:** Summary of the K_m_ and V_max_ values of plant, algal, cyanobacterial, and ancestral GOX-like proteins. The table displays mean values and standard deviation of the V_max_ and K_m_ values with the substrates glycolate or l-lactate from measurements with at least three biological replicates. The enzymes are ordered as inferred from the phylogenetic tree displayed in Figure 2, that is, starting with the hypothetical ancestral form N3-GOX, followed by the cyanobacterial (No-LOX) and chlorophytic LOX (Cr-LOX) enzymes, and then the GOX proteins from Archaeplastida, namely: glaucophytic (Cp-GOXc), rhodophytic (Cm-GOX), and streptophytic (Sp-GOX) toward plant (At-GOX2) enzymes.

Enzyme	Organism	Reference	l-Lactate		Glycolate	
			V_max_ (µmol min^−1^ mg^−1^)	K_m_ (mM)	V_max_ (µmol min^−1^ mg^−1^)	K_m_ (mM)
N3-GOX	Synthetic ancestral protein	This study	0.17 ± 0.01	13.73 ± 1.93	0.87 ± 0.05	11.8 ± 0.61
No-LOX ^x^	*Nostoc* sp. PCC 7120	Hackenberg et al.^x^	12.73 ± 1.55	0.04 ± 0.01	0.05 ± 0.02	0.23 ± 0.05
Cr-LOX ^x^	*Chlamydomonas reinhardtii*	Hackenberg et al.^x^	10.59 ± 0.46	0.08 ± 0.03	0.19 ± 0.06	1.24 ± 0.06
Cp-GOXc	*Cyanophora paradoxa*	This study	6.94 ± 0.76	9.27 ± 1.44	8.98 ± 1.25	0.38 ± 0.05
Cm-GOX ^y^	*Cyanidioschyzon merolae*	Rademacher et al.^y^	3.27 ± 0.35	14.92 ± 2.99	1.6 ± 0.29	0.9 ± 0.23
Sp-GOX	*Spirogyra pratensis*	This study	24.21 ± 6.73	15.52 ± 4.55	28.09 ± 2.78	0.94 ± 0.12
At-GOX2 ^x^	*Arabidopsis thaliana*	Hackenberg et al.^x^	0.74 ± 0.04	0.36 ± 0.18	35.64 ± 11.16	1.91 ± 0.64

^x^ [30]; ^y^ [38].

**Table 2 plants-09-00106-t002:** Amino acid positions of GOX-like proteins responsible for l-lactate or glycolate specificity. The proteins from the cyanobacterium *Nostoc* and the green algae *Chlamydomonas* exhibit amino acids shown to determine for l-lactate specificity, whereas the corresponding three amino acids of the ancestral (N3-GOX), glaucophytic (Cp-GOXc), rhodophytic (Cm-GOX), streptophytic (Sp-GOX) and plant (At-GOX2) protein, determine for glycolate specificity [30]. M—methionine; L—leucine; F—phenylalanine; T—threonine; W—tryptophan; V—valine.

	Amino Acid Position in No-LOX		
	82	112	212
**l-Lactate oxidases**			
No-LOX	M	L	F
Cr-LOX	M	V	F
**Glycolate oxidases**			
N3-GOX	T	W	V
Cp-GOXc	T	W	V
Cm-GOX	T	W	V
Sp-GOX	T	W	V
At-GOX2	T	W	V

**Table 3 plants-09-00106-t003:** Sequence similarities (in %) between the ancestral oxidase (N3-GOX), the plant glycolate oxidase (At-GOX2), and the cyanobacterial l-lactate oxidase (No-LOX).

	N3-GOX	No-LOX	At-GOX2
**N3-GOX**	1	0.56	0.68
**No-LOX**	0.56	1	0.46
**At-GOX2**	0.68	0.46	1

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
