# Peer review of "Evolution of Photorespiratory Glycolate Oxidase among Archaeplastida"

_plants, 2020, doi:10.3390/plants9010106_

Round 1

Reviewer 1 Report

The manuscript Evolution of Photorespiratory Glycolate Oxidase among Archaeplastida describes an evolutionary analysis of GO genes across phyla. The authors go on to describe that GOX genes are controversial in their evolutionary origin and likely were a result of horizontal gene transfer before the endosymbiotic event that led to plastid evolution. Through phylogenetic analysis and characterization of enzyme activities the authors demonstrate the origin of GOX proteins from a ancestral cyanobacterial source. The manuscript is well written and though the authors begin the results section demonstrating that this is a reanalysis of GOX origin the added data and new interpretation is supported by biochemical characterization. 

The biochemical characterization of the synthetic ancestral N3-GOX shows activities well below the enzymes present in the tested organisms. Although this is a modeled predicted enzyme would you expect this enzyme to be sufficiently functional in an ancestral species? Could transformation of this gene into a cyanobacteria rescue a defective phenotype? It is mentioned in the results section that the N3-GOX enzyme has lower activity but it would be interesting to see if it could be functionally relevant or if further modeling would be required.  It is an interesting note that proto-algal species would likely have 2 GOX genes after the endosymbiotic event and that one would be lost in evolution. It would be of interest if any of the available data present could predict which gene was likely to  be maintained or to add to the discussion the if one gene might have been lost during the transition of plastid to nuclear gene transfer. In addition, would the current data show any evidence of when the evolution of GOX localization to the peroxisome would have occured? If data is available adding this to results would be of interest, if this is out of the scope of the experimental data added discussion points could also be of value. 

Reviewer 2 Report

This work is a good attempt at identification of evolutionary origin of glycolate oxidases. However, the following points need further attention:

Major comments:

64-77: I think that a broad (non-expert) audience would benefit from a figure that compares the glycolate metabolism pathways in higher plants and algae/cyanobacteria, clearly indicate the reaction catalyzed by glycolate oxidase and/or other glycolate processing enzymes.

377-388: move to the results section (where I think it was initially). It’s quite important that readers know on what basis the searches were performed. Refer to Table S1 in the main text. Also, explain clearly why At GOX2 was used and justify your choice with references. This point is especially important since based on the phenotype of gox1 cat2 and gox2 cat2 double mutants GOX2 is not the primary GOX in Arabidopsis. GOX1 and GOX2 differ by just a few amino acids so it’s not a big deal for the phylogenetic studies, however the choice should be justified.

Figure 2: I am curious about the origin of the data presented in this figure. In materials and methods I see cloning and protein purification steps for N3-GOX, Sp-GOX and Cp-GOX but nothing about the rest. The origin of this data should be clearly explained. Are these values taken from previous studies? If yes, they should not be presented in this figure.

Throughout the manuscript the authors attempt to determine substrate preferences (glycolate vs lactate) for the respective enzymes (which is crucial for the interpretation of the data). However, generally, Vmax or Km are not used for this purpose and instead kcat/Km ratio is utilized. If possible, please adapt the manuscript to fit the generally accepted methodology. While doing so, consider the following reference: https://www.ncbi.nlm.nih.gov/pubmed/17433847

Minor comments:

45: replace “but see” with a text that explains why this reference is important;

52: specify that this is the first reaction of the photorespiratory pathway;

57: state that 2PG is rapidly metabolized to glycolate;

59: rephrase to avoid the use of word “typically”;

115: ref to supplementary material

122 and throughout the manuscript: make sure that the numbering corresponds to the order of appearance in the manuscript text

Supplementary material 2: color code the amino acids so readers can cee the similarities easier. Indicate the amino acids that determine the glycolate specificity.

129 and throughout the manuscript: capitalize the taxa names

Supplementary Figure S2: explain the use of red color

Legend of Figure 1: provide references to original papers. Next to the proteins analyzed by others, indicate which proteins were biochemically characterized in this study.

Table1: Include a column with references.

Table 3: describe the units

Provide more details on the cloning process, make sure that Table S2 has all primers used in this study.

Round 2

Reviewer 2 Report

Most of my comments were addressed and after a few additional changes I will be able to recommend this work for publication.

Additional comments:

I think that the authors misunderstood my comment concerning the data presented in previous version of Figure 2. In my opinion, original data, and data published earlier elsewhere can not be presented on the same graphs. I suggest to completely remove the data already published elsewhere (Vmax and Km values obtained for No-LOX, Cr-LOX, Cm-GOX, At-GOX2) from the graphs of current Figure 5, and retain only values obtained for Cp-GOXc and Sp-GOX which truly belong to this paper. As it is now, it looks like the authors wish to publish the same data twice. I understand the intention of comparing the Vmax and Km values obtained for the respective enzymes and substrates, but for this purpose, referring to previously published data in Table 1 is absolutely sufficient. Once the previously published data are removed from current Figure 5, I suggest to move graphs describing Km and Vmax for Cp-GOXc and Sp-GOX into the upper panel of current Figure 3 (layout as in old Figure 2) or to present them in an independent figure. the correct units for kcat/Km are M−1 s−1 please correct the manuscript figures and supplementary material should be numbered in the order of appearance in the text
